# Analysis of Key Factors Affecting Low-Carbon Travel Behaviors of Urban Residents in Developing Countries: A Case Study in Zhenjiang, China

**Pengfei Zhao** [1] , **Lingxiang Wei** [2,3,\*] , **Dong Pan** [4] , **Jincheng Yang** [3] **and Yuchuan Ji** [3]

1   School of Civil and Transportation Engineering, Beijing University of Civil Engineering and Architecture, Beijing 102616, China
2   School of Defense Engineering, Army Engineering University of PLA, Nanjing 210007, China
3   School of Materials Science and Engineering, Yancheng Institute of Technology, Yancheng 224051, China
4   School of Automobile and Transportation and Engineering, Hefei University of Technology, Hefei 230009, China
\*   Correspondence: weilx@ycit.edu.cn

**Abstract:** The transport sector accounts for 23% of global carbon emissions, the second largest after electricity and heat generation. Low-carbon travel, such as walking, cycling, and use of public transit, has become an effective way to reduce transportation-related emissions, however, what factors and how they affect low-carbon travel behavior remain unclear. This paper analyzes the key factors that affect the low-carbon travel behavior of urban residents by exploring 26 potential correlation indicators. Low-carbon travel survey data of urban residents in Zhenjiang, China are used as an example. Five factors derived from 26 indicators were identified and named as key factors influencing urban low-carbon travel behavior: traveler personal attributes (contribution rate 31.646%), user satisfaction with travel processes (contribution rate 17.438%), attitude towards low-carbon travel and environmental awareness (16.090%), the level of public transportation system development (10.793%), and overall attributes of travelers' family (10.561%). The results provide a theoretical basis for the development and implementation of novel urban low-carbon travel concepts in the future.

**Keywords:** green transportation; low-carbon travel; travel behavior; information process; factor analysis

## 1. Introduction

In the context of recently defined carbon peak and carbon neutralization goals, low-carbon travel has become an efficient means of lowering carbon emissions, including the use of bus and subway systems, bicycles, and other public or electric vehicles for city transit [1,2]. Nevertheless, the transportation sector is responsible for 23% of global carbon emissions, making it the second-largest sector after electricity and heat generation [3–5]. This is primarily due to the widespread ownership and use of automobiles in many cities, especially in North America and Australia. Simultaneously, the Asia-Pacific region is undergoing rapid industrialization and urbanization, which has resulted in dramatic shifts in transportation demands over the past few decades [6,7]. The transportation sector continues to rely heavily on a steady and economical oil supply [8]. The transportation sector's global oil consumption is projected to quadruple by 2050 [9], primarily in emerging nations where growing urbanization and rising wealth have caused a significant expansion in the usage of private motorized transport [10,11]. Therefore, transportation sectors in developing countries should change toward cleaner, more sustainable, low-carbon transport growth routes [12,13].

Travel-induced emissions include the greenhouse gases (GHGs) that contribute to climate change, most notably, carbon dioxide ($CO_2$), as well air pollutants, such as nitrogen

oxides ($NO_x$) and primary particulate matter of less than 2.5 μm (PM 2.5), which degrade urban air quality [14,15]. This study uses "low-carbon" as an abbreviation for both low $CO_2$ emissions and low emissions of local air pollutants, both of which are predominantly produced from the burning of carbon-based fuels [14,16]. Globally, several travel demand management/mobility policies have been implemented to encourage passengers to pick low-carbon transport modes, with modest effectiveness in terms of mode changes or other aggregate travel volume implications [17].

Historically, the behavioral intention is crucial to actual behavior when evaluating such a selection approach [18], the value-belief-norm theory (VBN) and theories of planned behavior (TPB) have been used for traveler intentions toward low-carbon travel options [19–21]. The fields of pro-environmental behavior, such as energy consumption behavior, have validated both VBN and TPB [22] and travel behavior [23], including private car use [24]. Previous studies focused on extending TPB or VBN models by introducing variables such as descriptive norms [25], perceived environmental concerns [20,26], perceived moral obligation [23], green trust [27], and habit [28]. Low-carbon transportation solutions are also influenced by objective factors, such as comfort, cost, duration, and purpose [29,30]. Another important factor influencing low-carbon travel behavior is whether a resident is environmentally conscious. As a result, it is important to research if locals' environmental consciousness has any bearing on their propensity for low-carbon travel. However, there are other elements that may influence whether or not tourists opt for a low-carbon mode of transportation [31,32].

It is common practice to use factor analysis to investigate a high number of measured variables (survey items) with a limited number of underlying factors (latent variables) [33–35]. The latent variables are then employed in other analyses like cluster analysis or regression. Additionally, the validity of metrics can be evaluated using factor analysis (the extent to which the constructs represent the original variables) [36,37]. Since the bulk of travel behavior characteristics are not quantifiable, exploratory factor analysis has been used extensively in research on travel behavior. Rather, many indicators are used to measure different elements [38,39].

To date, studies have either focused on people's psychological states or car use behavior, however, both internal (e.g., psychological) factors [40] and external (e.g., impact of transport policy) factors [41,42] have not specifically taken a two-dimensional viewpoint into consideration. This study looks into the major variables influencing urban inhabitants' low-carbon transport habits. The primary indicators of urban low-carbon travel uptake are examined, and potential critical determinants are explored, using exploratory factor analysis of multivariate statistics. To start, the number of selected indicators of urban residents' low-carbon transport modes is reduced using exploratory factor analysis. Then, to identify the pertinent indicators, a case study based on survey data gathered through questionnaires is conducted. In the end, a mathematical model of the critical variables influencing urban low-carbon travel behavior is developed.

The reminder of this paper is organized as follows. Section 2 introduces previous research on low-carbon strategies and models, travel behaviors and its influencing factors, low-carbon travels. Section 3 describes methods to determine and analyze key factors affecting whether or not a resident chooses low-carbon travel. Section 4 presents the case study, including the data source, applicability test, and data extraction; then, the key factors are identified. The main findings are summarized and future research directions are discussed in Section 5.

## 2. Related Works

### 2.1. Low-Carbon Strategies and Models

Various studies analyze low-carbon strategies and models for supply chain emission reduction, including game approaches, differential models, and finance indices. Other studies investigate low-carbon investments and their value in portfolios, as well as methods for promoting low-carbon assets in financial institutions.

Bangkok's shift in transportation mode from private cars to public systems has significantly reduced energy requirements, carbon emissions, and local air pollutants [43]. The UK has studied strategic approaches to low-carbon transportation, and compared them to previous transport systems. By coordinating efforts between the government and the market, the aim is to use economic measures to control the growth and usage of private cars, improve the design of public transportation facilities, and increase comfort levels to reduce the usage rate of private cars. To implement this concept, the UK government established a bike lane in Bristol, making it the first "bicycle city" in the UK [44].

China launched a pilot low-carbon city project to explore a low-carbon path for 2010. The Long-Range Energy Alternatives Planning System (LEAP) model was used to simulate the pilot in Ningbo, and it is expected that carbon dioxide emissions will reach 6.5183 million tons by 2050. By combining the Energy System Optimization (ESO) scheme and the Policy-guided Energy Saving (PES) scheme, carbon dioxide emissions can be reduced by 27%. Various measures and multilateral actions are taken to achieve a carbon development path and to use it for future low-carbon trajectories [45].

Additionally, studies examine the impact of low-carbon policies on enterprise total factor productivity (TFP) and provide insights into low-carbon hydrogen production methods. Emidia et al. (2018) utilized survey data in Italy and employed a mixed methods analysis to identify factors affecting low-carbon mobility. They proposed a low-carbon transition management model that includes developing clean energy, increasing public transportation, and transforming the role of private cars [46]. Chew et al. (2017) argue that a low-carbon society is one of the mechanisms deployed to achieve green economic growth. By analyzing factors that affect global climate change and using mathematical statistics and relevant data, they believe that high carbon dioxide emissions are the main cause of climate change and that many Asian countries are highly dependent on non-renewable energy sources. They advocate for greater attention to green transportation and propose the integration of water, energy, and materials to promote low-carbon implementation [47].

### 2.2. Travel Behaviors and Its Influencing Factors

Various academic studies are related to travel behaviors. Researchers have used various methods, including robust random forest analysis, sequence analysis, and cross-lagged panel modeling, to analyze data from different regions and countries. For example, Cheng et al. (2019) proposed a robust random forest method to analyze travel mode choices for examining prediction capability and model interpretability [48]. Tang et al. (2019) defined app-based ride-hailing services to include hailing of taxis through smartphone apps and sharing of private vehicles (car sharing is not included) [49]. Mouratidis et al. (2019) examined how urban form affects travel satisfaction using survey and interview data from the Oslo metropolitan area, which is a good case for such a study since compact and sprawled urban forms are found to a large extent in the same city region [50]. Sequence analysis was also introduced to measure fragmentation in activity participation and travel, using this technique, McBride et al. (2020) studied the exploration of statewide fragmentation of activity and travel and a taxonomy of daily time use patterns in California, USA [51].

Many of these shifts were already underway for a long time, but the pandemic has accelerated them remarkably. Shamshiripour et al. (2020) endeavored to examine the degree and nature of changes in individuals' mobility patterns and customary travel behaviors during the COVID-19 pandemic. Furthermore, they sought to determine whether these alterations would persist post-pandemic or revert to conditions preceding the outbreak [52]. Given this pandemic situation, the major aim of Han et al.'s (2020) study was to develop a conceptual framework that clearly explains the US international tourists' post-pandemic travel behaviors by expanding the theory of planned behavior (TPB) [53]. Abdullah et al. (2020) found that gender, car ownership, employment status, travel distance, the primary purpose of traveling, and pandemic-related underlying factors during COVID-19 are significant predictors of mode choice [54]. Shakibaei et al. (2020) aimed to investigate the

impacts of the pandemic on travel behavior in Istanbul, Turkey, through a longitudinal panel study conducted in three phases during the early stages of the epidemic and pandemic [55]. Miao et al. (2021) intend to offer some foundational building blocks to better understand travel and tourism behaviors and to inform tourism theory and practice in the post-COVID era [56].

Overall, the findings suggest that factors such as gender, car ownership, employment status, travel distance, and pandemic-related underlying factors during COVID-19 significantly predict mode choice during the pandemic. Existing studies aim to better understand travel and tourism behaviors and to inform tourism theory and practice in the post-COVID era.

### 2.3. Low-Carbon Travels

There has been a significant amount of research on low-carbon transport. Studies have indicated that $CO_2$ emissions are the main source of GHGs, making up around 66% of all GHGs [47]. As a result, low-carbon solutions are required to cut GHG emissions. The carbon footprint of logistics supply chains can be lowered, as Tang et al. (2015) demonstrated, by reducing the frequency of shipments [57]. Yang et al. (2018) analyzed the travel patterns of Beijing residents by examining five factors that influence their travel behavior: resident travel demand characteristics, travel mode characteristics, socio-demographic characteristics, subjective attitudes and perceptions, and environmental characteristics. They proposed measures to improve public transportation services by controlling the total number of car license plates and implementing time-limited restrictions, in order to encourage residents to choose low-carbon transportation options [58]. A low-carbon urban-rural ecosystem proposed by Penazzi et al. (2019) effectively reduces carbon emissions due to traffic [59]. As a basis for low-carbon block design, Hou et al. (2019) evaluated the spatial distribution of carbon emissions by choosing four components from traffic survey data, including trip mode, journey time, purposes of travel, and travel frequency [60]. Geng et al. (2018) calculated the efficiency of low-carbon travel regulations using data from a random sample survey of five Chinese cities. The highest level of policy reaction was shown in policies promoting the expansion of public transportation [32]. Sun et al. (2017) calculated the total carbon monoxide (CO), carbon dioxide ($CO_2$), and hydrocarbon (HC) emissions of passenger vehicles in Shanghai after studying traffic carbon emissions at the microscopic, medium, and macroscopic levels [61]. To explore the travel habits of suburban residents, Salonen et al. (2014) combined map survey data with a sophisticated multi-modal path analysis and found that slower modes of transportation typically emit less $CO_2$ than faster ones [30]. To support low-carbon development, additional study on low-carbon travel is still required. In addition, understanding the elements that influence low-carbon travel is extremely important.

### 2.4. Summary

Based on existing research, it can be seen that there are more and more studies on low-carbon topics. There are almost successful cases of low-carbon transportation and low-carbon economy in every country, but the number is still limited. Many studies are based on the low-carbon concept, such as how to use low-carbon materials to achieve the best benefits, mainly in the low-carbon transformation aspect. To achieve low-carbon, it is important to optimize urban layout, increase public transportation, plan routes, and reduce carbon dioxide emissions, which is the main culprit of global warming, by combining changes in the environment. With the development of the transportation industry, many scholars have begun to think about new issues, namely, how to achieve low-carbon fundamentally. Apart from inevitable freight transportation, the choice of residents' travel behavior accounts for a relatively important part, and investigation of residents' travel behavior is inevitable. Investigation methods are constantly improving, but there is less research on data application, and there are deviations in the investigation results due to the difficulty in collecting data. In recent years, the application of stated preference (SP)

and revealed preference (RP) in the investigation of residents' travel behavior has become more and more widespread, and many transportation researchers are constantly improving investigation methods. Through the analysis of residents' travel behavior choices, the main factors influencing residents' travel mode choices are the required costs, comfort, time duration, and habits. In addition, many classic models have been established and continuously improved, such as the common Logit model, Random Utility Maximization (RUM) model, aggregate model, non-aggregate model, and MNL (Multinomial Logit) model.

## 3. Methods

### 3.1. Analysis of Factors Influencing Low-Carbon Travel Behaviors

People are the main source of travel, and the mode of transportation they choose will be influenced by both subjective norms and objective factors, such as the types of transportation options that are available in a city, which can have a significant impact on whether urban residents choose low-carbon transportation. In addition, variables including mobility, environmental awareness, and the environment of urban traffic have an impact on low-carbon travel options [62].

(1)    Personal attributes and travel conditions

Residents' travel patterns can be influenced by a number of variables, including age, gender, educational background, occupation, and socioeconomic status. Elderly people, for instance, frequently travel shorter distances than young people, and their travel objectives are also very different. Academic standing may have an impact on a resident's perception of various means of transportation. The tendency to pick low-carbon vehicles is typically stronger among persons with greater educational backgrounds. Furthermore, factors that affect travel, such as comfort, cost, duration, destinations, and other factors, might objectively affect a resident's decision to choose a low-carbon mode of transportation. The travel index classification system for considering the personal attributes of residents and travel conditions is presented in Table 1.

**Table 1.** Indicator classification based on personal attributes and travel conditions.

| Indicators | Indicator ID |
|:---:|:---:|
| Age | X1 |
| Academic qualifications | X2 |
| Income | X3 |
| Occupations | X4 |
| Gender | X5 |
| Residence | X6 |
| Travel distance | X7 |
| Travel time | X8 |
| Travel comfort | X9 |
| Travel safety | X10 |
| Travel cost | X11 |
| Purpose of travel | X12 |
| Punctuality of transportation | X13 |
| Car ownership | X14 |
| Family members | X15 |

(2)    Environmental awareness and attitude of traveler

Another crucial element affecting low-carbon travel is the traveler's environmental consciousness. Investigated is whether a resident's environmental consciousness influences their propensity for low-carbon travel. A resident's opinion of the current state of the environment will determine how conscious they are of the environment. For instance, a traveler's attitude toward low-carbon travel will be more favorable if they believe environmental problems are getting worse. In contrast, if someone believes that environmental issues do not constitute a threat that is immediately present, they may view green travel

behavior unnecessary and have a less favorable opinion of low-carbon travel. The travel index classification system for considering the environmental awareness of residents is presented in Table 2.

**Table 2.** Indicator classification based on environmental awareness and attitude of traveler.

| Indicators | Indicator ID |
|---|---|
| Whether an individual believes low-carbon travel can solve environmental problems | X16 |
| Influence of environmental awareness of the people around | X17 |
| Whether an individual believes car use has an impact on the environment | X18 |
| Knowledge of garbage classification | X19 |
| Whether the garbage will be classified | X20 |
| Whether an individual believes cycling is an environmentally protective way to travel | X21 |

(3) Traffic environment and policies of the city

Additionally, significant influences on low-carbon travel behavior are the city's traffic regulations and laws. The suitable mode of transportation will be selected by residents based on their individual demands, and the mode of transportation also affects the carbon footprint. Residents frequently opt for public transit over alternative forms of mobility in cities with well-developed public transportation systems. Therefore, it is crucial to take into account how urban traffic patterns and policies affect low-carbon transport. The travel index classification system for considering traffic environment and low-carbon policies of a city are presented in Table 3.

**Table 3.** Indicator classification based on environmental awareness and attitude of traveler.

| Indicators | Indicator ID |
|---|---|
| Level of urban public transportation development | X22 |
| Purchase restrictions and car usage cost | X23 |
| Popularity of shared bicycles | X24 |
| Increase in taxi fare | X25 |
| Impact of government communications promoting low-carbon travel | X26 |

Based on the previous investigation, a system of essential elements influencing the adoption of low-carbon travel alternatives by city people was built using three basic factors. After that, 26 evaluation indicators were chosen and sorted into three categories. In conclusion, whether urban people choose low-carbon transportation options depends on the interaction of variables and their mutual influence.

*3.2. Method for Determining Key Factors*

The number of potential indications of whether urban residents will pick a low-carbon mode of transportation can be decreased using exploratory factor analysis [63,64]. First, survey data are obtained through questionnaires, then processed and analyzed. Then, the original set of indicators is reduced to a smaller number of relevant evaluation indicators in order to obtain the key factors affecting whether or not a resident will choose a low-carbon form of travel. Finally, a mathematical model of key factors affecting the adoption of urban low-carbon travel is constructed. Assuming a sample consists of the low-carbon travel behavior of all urban residents within a certain area, the 26 indicators are observed in each sample. The matrix of a sample with $n$ data points can be calculated by Equations (1) and (2).

$$X = \begin{bmatrix} X_{11} & X_{12} & \cdots & X_{1,26} \\ X_{21} & X_{22} & \cdots & X_{2,26} \\ \vdots & \vdots & \ddots & \vdots \\ X_{n1} & X_{n2} & \cdots & X_{n,26} \end{bmatrix} \tag{1}$$

$$x_j = \begin{bmatrix} x_{1j} \\ x_{2j} \\ \vdots \\ x_{nj} \end{bmatrix}, j = 1, 2, \cdots, 26 \tag{2}$$

Key factor analysis of low-carbon travel affecting urban residents is then used to synthesize the 26 observation indexes into a new comprehensive index of $m$ ($m < 26$) key factors, that is

$$\begin{cases} X_1 = a_{11}F_1 + a_{12}F_2 + \cdots + a_{1m}F_m + \varepsilon_1 \\ X_2 = a_{21}F_1 + a_{22}F_2 + \cdots + a_{2m}F_m + \varepsilon_2 \\ \cdots \\ X_{26} = a_{26,1}F_1 + a_{26,2}F_2 + \cdots + a_{26,m}F_m + \varepsilon_p \end{cases} \tag{3}$$

where $F$ is the key factor of low-carbon travel index of urban residents, $\varepsilon$ is the special factor of the low-carbon travel index ($X$) of each urban resident, $a_{ij}$ is the factor load, and the matrix composed of all $a_{ij}$ is the low-carbon travel factor load matrix of urban residents.

Requirements of low-carbon travel index ($X$) are as follows: (1) $F_i$ and $F_j$ are related and have a complementary relationship ($i \neq j$) with the same variance of 1, $m \leq p$; (2) $F$ and $\varepsilon$ are unrelated but have a complementary relationship with different variances.

*3.3. Key Factor Analysis*

The key factors for modeling low-carbon travel behavior among urban residents were determined using principal component analysis. This method has been widely successful in the fields of economics and engineering, as detailed in reference [65]. The method can be divided into six main steps, as follows:

1.  The original 26 indicators of low-carbon travel behavior are selected, standardized, and converted into the required data types;
2.  A correlation coefficient matrix consisting of the 26 indicators is established;
3.  The eigenvalue of the correlation coefficient matrix and the corresponding unit eigenvector are obtained, and the previous eigenvalue and the corresponding eigenvector write factor load matrix are selected according to the size of the cumulative contribution rate;
4.  The matrix of the factor load is subjected to the maximum orthogonal rotation of the variance;
5.  Factor scores of low-carbon travel behavior of residents are calculated, the coefficient matrix of factor component scores is output, and the model of key factors affecting low-carbon travel of urban residents is established;
6.  Based on the calculated low-carbon travel factor scores, the key factors affecting low-carbon travel of urban residents are identified.

Assuming that after the factor analysis, the key factors of low-carbon travel of $k$ urban residents are transformed using the original 26 factor indicators, then the $k$ key factors can be expressed as

$$F_i = u_{i1}x_1 + u_{i2}x_2 + \cdots + u_{i,26}x_{26} \ (i = 1, 2, \cdots, k) \tag{4}$$

where $F$ is the key factor of the low-carbon travel index ($X$) of an urban resident, and $u_{ij}$ is the factor score coefficient.

**4. Case Study**

Here, we selected Zhenjiang, China as our research area and analyzed the travel behavior characteristics of its residents using questionnaire survey data. Zhenjiang is a prefecture-level city in the eastern Chinese province of Jiangsu, see Figure 1. With a population of over 3.2 million, it is one of the fastest-growing cities in the region. Zhenjiang has a well-developed transport infrastructure that includes highways, railways, and waterways.

In terms of transportation, Zhenjiang has a variety of transport modes available to its residents, including buses, taxis, bicycles, and private cars. The city has also invested in a public bike-sharing system, which has become increasingly popular among commuters.

The travel habits of Zhenjiang residents are influenced by factors such as the availability and convenience of transport modes, population density, and the built environment. Many residents prefer to use private cars due to their convenience, but this has also contributed to traffic congestion and air pollution. The government has implemented policies to encourage the use of public transport and limit the growth of private cars, such as license plate restrictions and the development of more efficient and accessible public transport options.

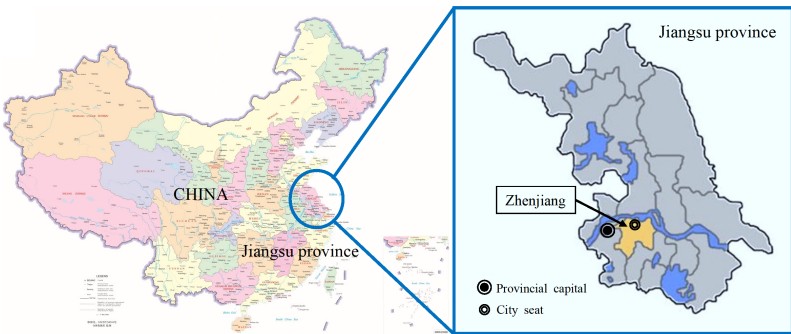

**Figure 1.** The location of Jiangsu province within China and the location of Zhenjiang City in Jiangsu province.

Survey data were obtained from OpenITS (http://www.openits.cn/, accessed on 9 January 2023), an online database of publicly available development data. The questionnaires were in the form of multiple-choice questions (a total of 45 questions) divided into three categories: travel information (12 questions), environmental awareness (4 choice questions), and attitude towards low-carbon travel (29 questions). The travel information survey mainly included questions related to family and work life; the environmental awareness survey was mainly used to obtain information on waste sorting and car use; the attitude towards low-carbon travel questions were devised to reflect traveler preferences for various modes of transportation. Respondents included residents of Zhenjiang and other parts of the Yangtze River Delta, such as Shanghai, Wuxi, Suzhou, and Changzhou. Based on the 45 multiple-choice questions, this paper screens for problems corresponding to the 26-indicator systems constructed above.

The survey area is within reach of the Zhenjiang high-speed railway station, an old railway station, and the long-distance bus station in Jiangsu Province. All three survey sites are located in the center of Zhenjiang City, where there is a large flow of people. Survey respondents had a wide range of occupations and academic qualifications. A mobile phone was used to directly submit answers. The time spent on each questionnaire was about 5 min and surveys were performed from 22 June to 26 June 2015. The total sample size was 2941, of which approximately 281 were invalid.

According to the survey results, the male-to-female ratio of the respondents was 1.1:1, which is consistent with the current male-to-female ratio of 1.06:1 among urban residents in Zhenjiang. In terms of occupational distribution, the occupational distribution of urban residents in Zhenjiang is as follows: Industry and construction workers: 23.4%; Service industry workers: 49.7%; Agricultural, forestry, animal husbandry, and fishery workers: 1.5%; Production, transportation, and operating workers: 19.3%; Others: 6.1%.

The income distribution of the respondents is shown in Figure 2.

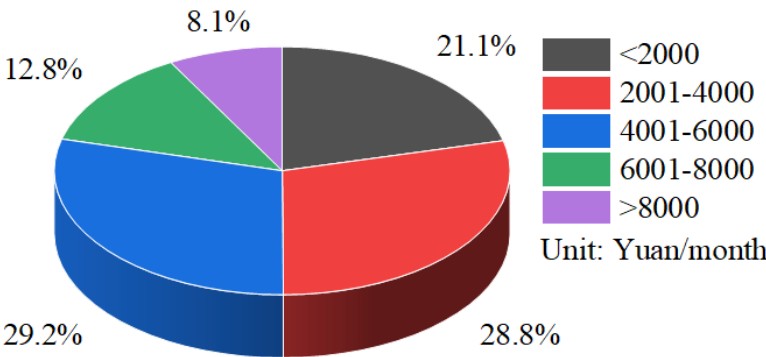

**Figure 2.** Income distribution of survey respondents.

*4.1. Applicability Test*

Table 4 shows the KMO and Bartlett's test results of the observed variables. An analysis of the model applicability was performed with SPSS Version 26.0 statistic software package [66]. The Bartlett sphere test value is 66,612.293 with 253 degrees of freedom and a *p*-value of 0.000 (<0.001), showing that the data meet the requirements of factor analysis. The KMO of the variable is 0.801 (>0.8), also indicating that the variable is suitable for factor analysis.

**Table 4.** KMO and Bartlett's test results.

| Validity Test Methods | | Values |
|---|---|---|
| Kaiser–Meyer–Olkin measure of sampling adequacy | | 0.801 |
| | Chi-square approximation | 6661.293 |
| Bartlett's test of sphericity | df | 253 |
| | Sig. | 0.0000 |

*4.2. Extraction of Key Factors*

After confirming that factor analysis can be performed, the principal component extraction method can be used to extract the key factors of 26 variables. Table 5 gives the variance explanations of the five key factors. From largest to smallest, the first principal component eigenvalue is 4.240 and the last is 0.689. The first four components presented in Table 5 have eigenvalues greater than 1 and the cumulative variance contribution rate reaches 86.527% or more. The five common factors can replace 26 of the variables of the original data. An orthogonal rotation was performed on the factor load matrix to maximize the variance. The cumulative variance after rotation was 86.527%, which is greater than 85%, in accordance with common factor description rules.

**Table 5.** The variance of explanation of principal component eigenvalue.

| Ingredients | Total | Initial Eigenvalue Variance % | Accumulation % | Total | Rotating the Sum of Squares Variance % | Accumulation % |
|---|---|---|---|---|---|---|
| 1 | 4.240 | 40.437 | 40.437 | 3.599 | 31.646 | 31.646 |
| 2 | 2.747 | 23.942 | 64.379 | 1.941 | 17.438 | 49.084 |
| 3 | 1.527 | 12.638 | 77.016 | 1.861 | 16.090 | 65.174 |
| 4 | 1.040 | 7.085 | 84.102 | 1.562 | 10.793 | 75.967 |
| 5 | 0.689 | 2.426 | 86.527 | 1.543 | 10.561 | 86.527 |

Figure 3 presents the scree plot of the factor analysis. The X-axis and Y-axis represent the component number and variances, respectively. It can be found that a typical gravel map has a distinct inflection point and a steep line connecting the key factors before the inflection point, followed by a line with a gentle slope connecting the small factors. In

addition, the eigenvalues of the top five factors are large and the key factors can be obtained according to the importance degree of the steepness judgment factor.

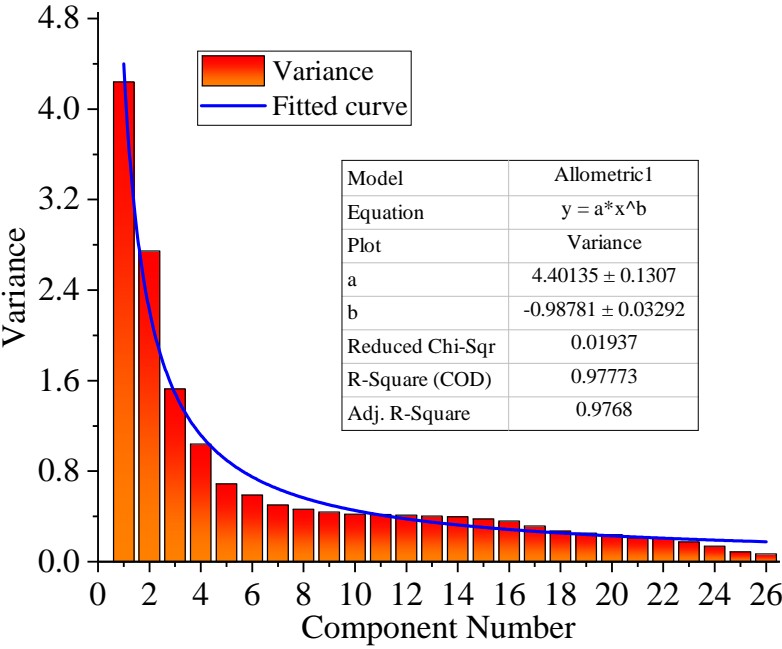

**Figure 3.** The scree plot of the factor analysis.

### 4.3. Identification of Key Factors

To achieve the desired dimensional reduction, the first five common factors with a cumulative variance contribution rate of more than 85% were extracted to replace all variables in the original data. Then, an orthogonal rotation was applied to the factor load matrix. The coefficient matrix of component scores after rotation is listed in Table 6. The composition conversion matrix is presented in Table 7.

**Table 6.** Component score coefficient matrix.

| Component | 1 | 2 | 3 | 4 | 5 |
|---|---|---|---|---|---|
| X1 | 0.209 | −0.089 | 0.037 | 0.067 | −0.051 |
| X2 | 0.019 | 0.002 | 0.088 | 0.018 | −0.037 |
| X3 | 0.219 | −0.027 | 0.148 | 0.143 | −0.099 |
| X4 | 0.187 | −0.063 | 0.199 | −0.155 | −0.323 |
| X5 | −0.021 | 0.025 | 0.029 | 0.337 | −0.137 |
| X7 | −0.079 | 0.022 | 0.013 | 0.005 | 0.018 |
| X8 | 0.142 | 0.024 | −0.215 | −0.145 | 0.121 |
| X9 | 0.202 | −0.005 | 0.143 | 0.078 | −0.147 |
| X10 | 0.001 | −0.049 | −0.187 | 0.113 | 0.018 |
| X11 | 0.016 | 0.634 | 0.069 | −0.035 | −0.068 |
| X16 | 0.013 | −0.183 | −0.319 | −0.213 | 0.151 |
| X17 | 0.028 | −0.169 | 0.075 | 0.158 | 0.129 |
| X18 | 0.564 | 0.059 | −0.224 | 0.024 | 0.125 |
| X19 | 0.015 | 0.122 | 0.134 | −0.046 | −0.048 |
| X20 | −0.135 | −0.022 | −0.201 | 0.154 | 0.111 |
| X22 | −0.052 | 0.117 | 0.163 | 0.251 | 0.055 |
| X23 | 0.061 | 0.219 | 0.275 | −0.214 | 0.072 |
| X24 | −0.017 | −0.067 | 0.102 | 0.078 | 0.096 |
| X14 | 0.141 | −0.194 | 0.008 | 0.073 | 0.055 |
| X15 | 0.183 | 0.171 | −0.037 | −0.033 | 0.072 |

Based on the results shown in Table 7, the five key factors affecting low-carbon travel can be identified. The first set of key factor indicators—X1, X2, X3, X4, and X5—have load factors above 0.80 and reflect the influence of traveler personal attributes on low-

carbon travel behavior, therefore, "traveler personal attribute" is named as a key factor. The second set of key factor indicators—X7, X8, X9, X10, and X11—also have load factors above 0.80 and reflect the influence of satisfaction with the travel process on low-carbon travel behavior, therefore, "satisfaction with the travel process" is named as the second key factor. The third set of key factor indicators—X16, X17, X18, X19, and X20—have load factors above 0.80, and reflect the influence of traveler attitude toward low-carbon travel and environmental awareness on low-carbon travel behavior, therefore, "attitude toward low-carbon travel and environmental awareness" is identified as a key factor. The fourth set of key factor indicators—X16, X17, X18, X19, and X20—have load factors above 0.80 and represent the influence of public transportation system development and low-carbon policy guidance on low-carbon travel behavior, therefore, "the level of public transportation system development" is identified as a key factor. The fifth set of key factor indicators—X14 and X15—have load factors above 0.80, and represent the influence of the overall attributes of a traveler's family on low-carbon travel behavior, therefore, "overall attribute of traveler's family" is named as the final key factor.

**Table 7.** Compositional transformation matrix.

| Component | 1 | 2 | 3 | 4 | 5 |
|---|---|---|---|---|---|
| X1 | 0.875 | 0.413 | 0.038 | 0.027 | 0.087 |
| X2 | 0.859 | −0.214 | 0.044 | −0.302 | 0.020 |
| X3 | 0.845 | 0.251 | 0.300 | 0.078 | 0.050 |
| X4 | 0.843 | 0.402 | 0.331 | 0.351 | 0.050 |
| X5 | 0.818 | 0.099 | −0.001 | −0.371 | −0.118 |
| X7 | 0.107 | 0.898 | 0.495 | 0.459 | −0.408 |
| X8 | −0.072 | 0.875 | −0.174 | 0.220 | 0.032 |
| X9 | −0.064 | 0.864 | 0.322 | 0.021 | −0.143 |
| X10 | 0.124 | 0.890 | 0.212 | 0.034 | −0.215 |
| X11 | 0.332 | 0.841 | −0.385 | −0.213 | 0.132 |
| X16 | −0.021 | 0.015 | 0.876 | −0.118 | 0.136 |
| X17 | 0.345 | −0.067 | 0.899 | 0.212 | 0.328 |
| X18 | 0.242 | −0.376 | 0.826 | 0.021 | −0.131 |
| X19 | 0.145 | 0.377 | 0.873 | −0.343 | −0.009 |
| X20 | 0.270 | 0.056 | 0.819 | −0.112 | 0.068 |
| X22 | 0.222 | 0.062 | 0.118 | 0.878 | −0.309 |
| X23 | −0.123 | −0.033 | −0.214 | 0.889 | −0.256 |
| X24 | −0.409 | −0.089 | 0.312 | 0.832 | 0.221 |
| X14 | −0.253 | −0.046 | −0.371 | 0.078 | 0.877 |
| X15 | −0.467 | 0.564 | −0.225 | 0.044 | 0.887 |

## 5. Results and Discussion

### 5.1. The Relationship between Personal Characteristics and Low-Carbon Travel

In terms of the number of family members, the results of the travel survey showed that the number of family members did not have a significant impact on the choice of transportation mode. This may be because the low-carbon travel awareness of most respondents is relatively weak. Regardless of the number of family members, when all family members travel together, they still prefer to choose private cars with higher comfort as the main mode of transportation, rather than choosing low-carbon public transportation.

In terms of the composition of travelers, the composition of travelers has a significant impact on the choice of transportation mode. This may be because the travel purpose differs for different types of travelers. For instance, young couples usually prefer walking or taking public transportation for leisurely walks or to use existing tools when traveling together. Families with children or elderly members, on the other hand, tend to choose private cars for travel when the distance is moderate, as private cars are more convenient than low-carbon public transportation.

In terms of the gender of travelers, there is no significant difference in the number of trips between different genders, but females tend to travel slightly more than males. This may be due to the domestic responsibilities that females tend to shoulder, such as shopping and chauffeuring children to school, which are usually completed by women.

Moreover, there is a stronger correlation between women and low-carbon travel, while men tend to have a slightly lower correlation. This may be because men prioritize their psychological satisfaction, such as the feeling of status that driving a car brings, more than women do.

Age, income, occupation, and transportation mode are closely related. The elderly and children tend to choose low-carbon public transportation, which may be due to the physical impact that the elderly and children face. Residents with average income tend to emphasize the economy of travel and compare relatively comfortable and economical commuting methods, such as low-carbon public transportation. However, residents with higher income tend to choose convenient and comfortable modes of transportation, and are more inclined to buy cars. The satisfaction brought by others' recognition is also a factor that prompts residents to choose private cars. Travelers with more fixed occupations tend to travel less frequently than other travelers.

There is a significant correlation between education level and resident travel. Most highly educated individuals have a stronger sense of responsibility and consider the impact of their transportation choices on themselves and others. They are fundamentally motivated by this sense of responsibility to choose environmentally friendly modes of transportation and do not stop their efforts even if they do not see any immediate benefits to themselves from their choices.

Whether or not one owns a private car is an important factor affecting residents' low-carbon travel. People who own private cars tend to prefer using their own vehicles when traveling.

*5.2. The Relationship between Travel Characteristics and Low-Carbon Travel*

There is a significant correlation between travel time and transportation mode. With the accelerating pace of life, travelers are paying more attention to travel time. Changes in travel time can lead to changes in travel decisions, and the ease of changing travel decisions determines the mode of transportation used during travel. Individuals who value time tend to choose transportation with relatively accurate schedules, such as rail transit, while during rush hours, travelers tend to choose public transportation under the priority policy. During non-congested times, private cars are preferred by this type of traveler as they require less travel time.

*5.3. The Relationship between Other Attributes and Low-Carbon Travel among Residents*

There is a significant correlation between comfort and transportation mode, which is a key factor affecting residents' low-carbon commuting choices. Most residents are willing to choose private cars for travel as they provide greater comfort.

Safety, on the other hand, did not show a significant correlation with transportation mode. This may be because respondents believe that there is no significant difference in safety among the various transportation modes available currently.

The results also indicate that parking difficulties and road congestion are significant factors that affect the use of private cars. The lack of parking facilities is a concern for residents who use private cars for travel. They are unable to find parking spots and are subject to parking tickets, which are factors to be considered when using private cars for travel. Residents also consider whether the destination has parking spots or nearby convenient parking spots.

Due to factors such as congestion during rush hour and the urgency of time, residents generally choose more punctual modes of transportation, such as rail transit.

In addition, there is a significant correlation between traffic accidents and the choice of transportation mode, but traffic accidents are unpredictable and generally only regretted after an individual has experienced one. Therefore, the impact of traffic accidents on transportation mode is only realized after the fact.

Travel costs also have a significant impact on the choice of transportation mode, which generally depends on the distance and mode of transportation. This suggests that to promote low-carbon travel, the cost of travel should be reduced.

## 6. Conclusions and Future Research Directions

Taking low-carbon travel survey data of urban residents in Zhenjiang as an example, this paper comprehensively analyzed the key factors influencing the low-carbon travel behavior of urban residents using the exploratory factor analysis of multivariate statistics. In contrast with other studies, this paper considered the impact of both internal and external factors on low-carbon travel behavior and examined 26 potential correlation indicators to define the key factors. Five key factors influencing urban low-carbon travel behavior were extracted and named. The results can be used to guide future management decisions of urban low-carbon travel development.

According to the results of Table 5, the key factors affecting urban low-carbon travel behavior are traveler personal attributes (contribution rate of 31.646%), satisfaction with the travel process (17.438%), attitude towards low-carbon travel and environmental awareness (16.090%), level of public transportation system development (10.793%), and overall attributes of the traveler's family (10.561%). To a certain extent, these five key factors represent the main factors influencing the urban low-carbon travel behavior of residents. Thus, the findings provide a theoretical basis for developing and implementing future urban low-carbon travel concepts.

Based on the above results, there are several recommendations that could promote low-carbon travel in Zhenjiang city: (1) Encouraging the use of public transportation: Given the growing population and the increasing traffic congestion in the city, the government could promote the use of public transportation by improving the accessibility and convenience of buses and subways. This could include increasing the frequency of services, improving the infrastructure of bus stops and subway stations, and offering discounted fares to regular users; (2) Promoting active transportation: Zhenjiang's flat terrain and well-developed cycling infrastructure make it an ideal city for promoting cycling and walking. The city could invest in expanding its network of cycling lanes, providing secure bicycle parking, and promoting cycling as a safe, healthy, and sustainable mode of transportation; (3) Managing private car usage: Zhenjiang could adopt policies to limit the growth of private car usage, such as restricting the number of car licenses issued, increasing the cost of car ownership through taxes and fees, promoting carpooling and ride-sharing services; (4) Encouraging low-carbon commuting: Zhenjiang could encourage low-carbon commuting options by offering incentives for employees who choose to walk, cycle, or use public transportation to get to work. This could include offering subsidies for public transportation passes, providing secure bike parking facilities, and offering facilities for showering and changing at the workplace; (5) Promoting sustainable tourism: As a tourist destination, Zhenjiang could promote low-carbon travel options by developing eco-tourism activities and promoting sustainable tourism practices. This could include developing cycling and walking tours, promoting locally produced and sustainable food options, and encouraging visitors to use public transportation and active transportation modes.

Nevertheless, this paper is not without limitations. We only used data from Zhenjiang of China in our example analysis, therefore, the findings cannot be considered universal. In order to encourage low-carbon transportation behavior, research should also explore ways to integrate emerging technologies with existing transportation infrastructure. This could include the use of electric vehicles, ridesharing services, and intelligent transportation systems. Additionally, research should investigate how to optimize existing transportation networks to reduce the environmental impact of urban transportation. Travelers in different regions have different cultures, which will also have a certain impact on the behavior of travelers. Therefore, collecting travel information of travelers from different countries or regions will help to improve the analysis framework. In addition, different urban development levels and urban built environments are also extremely important, and how

to consider the impact of these factors on low-carbon travel choice behavior is also worth exploring. The next step will be to study key factors influencing the uptake of urban low-carbon travel in different regions and to identify the differences in key factors among different regions.

**Author Contributions:** Conceptualization, P.Z. and L.W.; Data curation, L.W.; Funding acquisition, P.Z.; Investigation, D.P., J.Y., and Y.J.; Methodology, P.Z. and L.W.; Validation, P.Z. and D.P.; Visualization, P.Z.; Writing—original draft, L.W. and D.P.; Writing—review and editing, P.Z. and L.W. All authors have read and agreed to the published version of the manuscript.

**Funding:** This study was supported by Beijing Postdoctoral Research Foundation (No. 2021-zz-111).

**Institutional Review Board Statement:** Not applicable.

**Informed Consent Statement:** Informed consent was obtained from all subjects involved in the study.

**Data Availability Statement:** Data will be made available on request.

**Acknowledgments:** The authors are very grateful to OpenITS for providing survey data (http://www.openits.cn/, accessed on 9 January 2023) for researchers.

**Conflicts of Interest:** All authors claim no conflict of interests in this study.

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
