# Peer review of "Analysis of Key Factors Affecting Low-Carbon Travel Behaviors of Urban Residents in Developing Countries: A Case Study in Zhenjiang, China"

_sustainability, doi:10.3390/su15065375_

Round 1

Reviewer 1 Report

I appreciate efforts that the Authors made to design and conduct their research and to prepare the paper. Please, follow the comments to improve the values of your article:

1. “…however, what factors and how they affect low-carbon travel behavior remain unclear” – there is a wide range of publications discussing findings of numerous studies on factors affecting transport behaviour, including sustainable mobility patterns. I suggest a necessary improvement in the text, namely developing the introduction and section 2 through a thorough and in-depth review of the literature on studies of transport behavior and behavioral change, both in terms of the meaning of different factors impacting this behaviour, efficiency of different policy measures aimed at required behavioral change, as well as sustainable travel behaviour as an element of sustainable consumption (including the application of TRA and TPB models). Such literature review would provide a solid background for the identification and selection of factors in section 3.

2. Is it a mistake in this sentence?: “This paper analyzed key factors affecting urban low-carbon travel behavior of residents by exploring 26 potential correlation indicators based on 26 potential correlation indicators.”. Moreover, it would be better to use present simple: “This paper analyses…”

3. In my opinion, a clear goal of the paper should be included in the introduction, not only a statement that “This study looks into the major variables influencing urban inhabitants’ low-carbon transport habits.” I am also wondering whether the Authors took into account that there can be locally- or regionally-specific differences in some factors impacting travel behaviour in China compared to other regions like e.g. European countries or the U.S.

4. As I mentioned in 1, the text should be augmented with publications from high profile journals – this is visible in section 3 as well which is poor in terms of a solid literature background for the identification of factors.

5. In section 4, the Authors should provide more information about Zhenjiang, especially in terms of various factors that can affect travel behavior of respondents (e.g. the level of the development of transport infrastructure, accessibility of different transport modes, travel habits, population size and density, built environment etc.).

6. The Authors should develop a discussion section to compare and evaluate their findings with other valuable studies.

7. In conclusion, the Authors should not only show the direction of future studies. The scientific and practical value added of their research should be underlined with more insights into possible application of the results (e.g. some policy recommendations for the required behavioral change).

Author Response

Thank you very much for your review, we have responded to your comments one by one. Please check it.

Reviewer 2 Report

This study is focusing on the key factors affecting low-carbon travel behaviors. As the environmental problem is being more serious, this topic is attractive. However, the meaning and academic contribution of this study are lacking.

          1. In particular, the paper only organized the factors into five categories, a deeper analysis such as how the factors affecting low-carbon travel behaviors are required.

          2.    Introduction:

-      The authors need to well improve the study’s originality and contribution. There are enormous previous studies focused on this topic. The differentiation of this study should be emphasized.

3.    Related works:

-      The related works need to be well organized. This section should be organized by the factors and methodologies presented in the previous studies. Also, the reason why the authors used those factors and methodology in this study.

4.    In the ‘Table 5. The variance of explanation of principal component eigenvalue.’, I wonder why the last ‘Accumulation %’ was not 100%?

5.    The study findings need to be improved well. According to the study’s result, what strategies should be implemented for low-carbon travel?

Author Response

(The authors gave the same response as above.)

Reviewer 3 Report

This paper analyzed factors that may affect low-carbon travel behaviors in urban areas. The study is interesting and has the potential to be used to encourage more low-carbon travel. However, the paper needs some major changes before it warrants publication. My specific comments are as follows:

More result analysis of the case study and its implications are needed. The paper is to analyze the key factors; however, the paper lacks the analysis of the factors identified in section 4.

In section 3.3, are the steps to model the key factor analysis a new method proposed in this paper? If not, where (e.g. reference) did these steps come from? From my point of view, these steps are similar to minimizing a sum of square errors in the least squares regression analysis. 

For the factors that are identified to be key factors that affect low-carbon travel behaviors in section 4, from my point of view, they may be some interaction or dependency. For example, a traveler’s attributes may affect his/her perception of the travel process and attitude. Could these potential interactions be explored?

For the case study, some summary statistics of the survey participants (e.g., age, gender, and occupation) are needed.

Author Response

(The authors gave the same response as above.)

Round 2

Reviewer 2 Report

This study was improved significantly. 

I have no more doubts.

Reviewer 3 Report

The authors have improved the paper and adequately addressed my previous comments. I have no more comments.